# Autoencoder-Based Extrasystole Detection and Modification of RRI Data for Precise Heart Rate Variability Analysis

**DOI:** 10.3390/s21093235

**Published:** 2021-05-07

**Authors:** Koichi Fujiwara, Shota Miyatani, Asuka Goda, Miho Miyajima, Tetsuo Sasano, Manabu Kano

**Affiliations:** 1Department of Material Process Engineering, Nagoya University, Furo-cho, Chikusa-ku, Nagoya, Aichi 464-8601, Japan; 2Department of Systems Science, Kyoto University, Kyoto 606-8501, Japan; miyatani.shota.73z@st.kyoto-u.ac.jp (S.M.); goda.asuka.67w@st.kyoto-u.ac.jp (A.G.); manabu@human.sys.i.kyoto-u.ac.jp (M.K.); 3Department of Liaison Psychiatry and Palliative Medicine, Tokyo Medical and Dental University, Tokyo 113-8510, Japan; miholppm@tmd.ac.jp (M.M.); sasano.bi@tmd.ac.jp (T.S.)

**Keywords:** heart rate variability analysis, extrasystole, RRI data, machine learning, autoencoder

## Abstract

Heart rate variability, which is the fluctuation of the R-R interval (RRI) in electrocardiograms (ECG), has been widely adopted for autonomous evaluation. Since the HRV features that are extracted from RRI data easily fluctuate when arrhythmia occurs, RRI data with arrhythmia need to be modified appropriately before HRV analysis. In this study, we consider two types of extrasystoles—premature ventricular contraction (PVC) and premature atrial contraction (PAC)—which are types of extrasystoles that occur every day, even in healthy persons who have no cardiovascular diseases. A unified framework for ectopic RRI detection and a modification algorithm that utilizes an autoencoder (AE) type of neural network is proposed. The proposed framework consists of extrasystole occurrence detection from the RRI data and modification, whose targets are PVC and PAC. The RRI data are monitored by means of the AE in real time in the detection phase, and a denoising autoencoder (DAE) modifies the ectopic RRI caused by the detected extrasystole. These are referred to as AE-based extrasystole detection (AED) and DAE-based extrasystole modification (DAEM), respectively. The proposed framework was applied to real RRI data with PVC and PAC. The result showed that AED achieved a sensitivity of 93% and a false positive rate of 0.08 times per hour. The root mean squared error of the modified RRI decreased to 31% in PVC and 73% in PAC from the original RRI data by DAEM. In addition, the proposed framework was validated through application to a clinical epileptic seizure problem, which showed that it correctly suppressed the false positives caused by PVC. Thus, the proposed framework can contribute to realizing accurate HRV-based health monitoring and medical sensing systems.

## 1. Introduction

The autonomous nerve system (ANS) regulates various physiological functions, such as circulation, respiration, digestion, sweating, thermoregulation, and metabolism, and is associated with various types of diseases [1]. Thus, disease diagnosis or symptom detection for clinical purposes in daily life would be possible with the realization of if real-time ANS activity monitoring.

A candidate solution is to use the heart rate variability (HRV), which is derived from an electrocardiogram (ECG). An ECG signal consists of peaks, such as the P, T waves and QRS complex, of which the highest peak is an R wave, and the interval between adjacent R waves is defined as an R-R interval (RRI) (ms). The heart rate variability (HRV) is a phenomenon in which there is fluctuation in the RRI, reflecting the activities of the ANS [2].

Although HRV analysis has been traditionally used in the cardiovascular field [3,4] and HRV analysis software have been developed [5,6], various new types of real-time health monitoring services have been developed based on HRV analysis. Drowsiness detection algorithms, which utilize the fact that HRV is altered due to the sleep stage transition [7], have been proposed [8,9,10,11]. Sleep apnea contributes to the development of cardiovascular events, which greatly affect HRV [12,13]. Research reported that apnea can be screened by monitoring HRV during sleep [14,15]. Epileptic seizures can be detected by means of HRV [16], since HRV changes during preictal phases [17,18]. In addition, an epileptic seizure prediction system was proposed, which combines HRV analysis and an anomaly detection algorithm [19].

To realize real-time health monitoring utilizing HRV, robust HRV feature extraction is essential; however, HRV features easily fluctuate when the raw data contains ectopic RRIs. For example, since the SDNN should be calculated from the RRI data measured during normal sinus states according to a clinical guideline for HRV analysis [2]. Ectopic RRIs must be detected and modified in real-time to calculate the SDNN. This problem should be solved on the software side after the ECG measurement; the RRI data collected from ECG sensors must be checked and appropriately treated by software before extracting the HRV when there is the possibility that ectopic RRIs are contained.

There are two major causes of an ectopic RRI: arrhythmia and R wave detection error due to ECG measurement failure or motion artifact contamination. Regarding the latter, electrode contact failure or sensor failure may cause long-term detection errors, as well as the inability to extract HRV features, as such failures inhibit the measurement of reliable RRI information. Although hazardous arrhythmia, such as long QT syndrome, may cause sudden cardiac death, persons without cardiovascular diseases also have certain types of nonhazardous arrhythmia that occur every day; for example, premature ventricular contraction (PVC) and premature atrial contraction (PAC) are the most common extrasystoles [20,21] in healthy persons.

This study focuses on only extrasystole treatment to improve the HRV analysis quality although there are various causes of ectopic RRI. In general, it is difficult to solve any type of problem by just one algorithm, which is sometimes known as the ‘no-free-lunch theorem’ in computer science [22]. Therefore, we adopted a ‘divide and conquer’ approach. In this study, extrasystoles are considered as they occur with low frequency in persons without any cardiovascular diseases because a person might have cardiovascular diseases when they have many extrasystoles [23].

Extrasystoles in the RRI data significantly affect the HRV analysis, causing performance deterioration of the ANS activity monitoring. Ectopic RRIs should be detected and modified appropriately before HRV analysis in order to improve the accuracy. Preferably, this ectopic RRI treatment is performed in real-time because drowsiness detection and epileptic seizure prediction are real-time applications. The simplest method for detecting extrasystoles may be the use of a threshold; however, such a strategy is not easily adopted. The heart rates of healthy adults typically range between about 50 to 90 bpm, that is, the range of RRI is about 650–1200 ms. On the other hand, the alternation width of the RRI by an extrasystole is about 300 ms. Thus, it is difficult to set an appropriate threshold for ectopic RRI detection that is applicable to all people.

Various ECG signal quality assessment methods have been proposed [24]. Kalkstein et al. proposed an erroneous signal detection method based on machine learning for ECG signals [25]. Jung and Kim proposed an extrasystole detection method from ECG signals using wavelet analysis [26]. Extrasystole detection based on ECG signals using a hidden Markov model (HMM) has been developed [27]. In addition, fuzzy neural networks or convolutional neural networks have been utilized for extrasystole detection from ECG signals [28,29].

However, it is desirable that extrasystoles are not detected from raw ECG signals but from the RRI data because dealing with ECG signals requires a much heavier computation burden compared with RRI data, although all of these methods analyze raw ECG signals. In fact, some wearable sensors do not measure the raw ECG signals but detect and collect only the RRIs for energy and computation savings for the embedded microcomputers [30]. Thus, we did not consider raw ECG signals but the RRI data for extrasystole treatment for precise HRV analysis in real-time.

The simplest treatment of ectopic RRI is to remove or ignore it [31]. The use of the Lomb–Scargle (LS) periodogram after ectopic RRI removal is recommended for frequency-domain HRV feature extraction [32]. However, such treatment is not preferable since it produces gaps in time relative to the real time, which is intolerant of real-time applications, such as epileptic seizure prediction systems or drowsiness detection. Mateo et al. analyzed the effect of ectopic beats on HRV [33] based on their proposed heart timing signal [34]. Kamata et al. proposed the use of locally-weighted partial least squares (LWPLS) for the interpolation of missing RRI caused by detection errors [35]. Their study showed that LWPLS could adequately interpolate missing RRIs and that it was difficult to modify more than two successive missing RRIs; however, Kamata’s method is unsatisfactory for our purposes because it does not mention extrasystoles.

To realize precise HRV analysis, we propose a new framework of nonhazardous extrasystole treatment utilizing an autoencoder (AE) [36] and a denoising autoencoder (DAE) [37], which are types of neural networks. The proposed framework consists of extrasystole occurrence detection from the RRI data and modification, where the targets are PVC and PAC. The RRI data are monitored by means of the AE in real-time in the detection phase, and the DAE modifies the ectopic RRI caused by the detected extrasystole. These are referred to as AE-based extrasystole detection (AED) and DAE-based extrasystole modification (DAEM).

The usefulness of the proposed framework was validated through its application to RRI data with artificial PVCs and PACs. The proposed framework performance cannot be evaluated by using real RRI data with extrasystoles since the “true” RRI values are unknown before the extrasystole occurs. Extrasystole detection by visual observation is burdensome, and  errors may occur even when cardiologists check real data, which makes it difficult to evaluate the precise performance of the proposed AED. In addition, the proposed framework was applied to a clinical epileptic seizure problem [19].

Although a preliminary version of this work was reported in [38], which proposed a PVC modification algorithm using DAEM, the detection of extrasystoles by AED, the modification of PAC using DAEM, and application to the real problem were not discussed therein.

## 2. Methods

### 2.1. Extrasystole

In this study, we consider PVC and PAC as these are the most common nonhazardous extrasystoles that occur even in healthy persons every day [20,21]. Other types of arrhythmia can be omitted because they rarely occur in healthy persons. This section introduces PVC and PVA and discusses their effects on HRV features. The HRV analysis method that is considered as an example in this work is described in the Appendix A.

#### 2.1.1. Premature Ventricular Contraction (PVC)

PVC is a common type of monostotic extrasystole, in which usually a heartbeat is skipped, as shown in Figure 1. A normal sinus rhythm is generated from the sinus nodes; however, cardiac activation sometimes originates from the ventricle, which causes a premature contraction [20].

Although PVC itself does not become a direct cause of death, PVC may induce ventricular fibrillation and lead to sudden death when PVC occurs in a patient with cardiovascular disease. Brodsky et al. also reported that at least one PVC occurred in 25 out of 50 men within 24 h of ECG monitoring [39]. According to a survey conducted by Kostis et al., 39 out of 101 healthy adults had a PVC at least once within 24 h [40]. These reports indicate that PVC is a common type of arrhythmia—occurring even in healthy persons. In particular, age plays a major role in the PVC occurrence throughout the population: under 1% of those affected are under the age of eleven years, and 6% are over 75 years old [41].

The occurrence of PVC alters the RRI and affects the HRV features. Figure 2 is a comparison of the SDNN and LH/HF derived from original RRI data and RRI data with an artificial PVC. These HRV features were significantly altered shortly after the PVC occurrence, and that the influence lasted for three minutes in this case, which is the window size of HRV calculation (three minutes). Since such discontinuous changes in signals can be regarded as very high-frequency components, the power of the HF region of the RRI data dramatically increases while that of the LF region does not greatly change. Accordingly, the LF/HF abruptly changes as shown in Figure 2.

#### 2.1.2. Premature Atrial Contraction (PAC)

PAC is also a common type of monostotic extrasystole characterized by premature heartbeats, as shown in Figure 3. Although the sinoatrial node regulates the heartbeat during the normal sinus rhythm, depolarization from other atrial regions before the sinoatrial node causes PAC [20].

PAC occurs even in healthy persons and usually does not require any attention. However, Jensen et al. reported that PAC preceded all cases of atrial fibrillation in patients [42]. Conen et al. reported that 99% of 1742 subjects aged 50 years or older without heart disease had at least one PAC during one-hour ECG monitoring [43]. They reported that the number of PAC occurrence increased with age. As with PVC, PAC affects HRV features.

### 2.2. Extrasystole Detection and Modification

This study proposes a new framework for detecting and modifying ectopic RRIs caused by extrasystoles for robust HRV analysis. The proposed extrasystole treatment framework utilizes an autoencoder (AE) and a denoising autoencoder (DAE), which are variations of neural networks.

#### 2.2.1. Autoencoder (AE) and Denoising Autoencoder (DAE)

An autoencoder (AE) is a type of neural network trained to output values as close as possible to the input variables, originally proposed as a method of dimensionality reduction and feature extraction [36,44]. The structure of an AE is illustrated in Figure 4.

In the training phase of AE, the cost function must decrease as the output variables become close to the input variables. In this work, we adopted an adjusted mean squared error function, which is expressed as follows:(1)$$\begin{array}{c} J=\frac{1}{N}\sum_{n=1}^{N}{∥x_{n}-{\hat{x}}_{n}∥}^{2}+\lambda \times \Omega_{L_{2}} \end{array}$$
where *N* is the number of samples, $x_{n}$ is an input vector of the *n*th (1≤n≤N) sample, and ${\hat{x}}_{n}$ is an output vector of the *n*th sample. λ is the coefficient for the $L_{2}$ regularization $\Omega_{L2}$.

AE can also be used for anomaly detection by using its reconstruction error RE:(2)$$\begin{array}{c} RE=||x-\hat{x}\left|\right| \end{array}$$
where *x* and $\hat{x}$ are the input variables and the output variables of the AE, respectively. To utilize an AE for anomaly detection, it has to be trained with only normal data. Since the AE is trained so that the output variables become close to the input variables in the normal condition, a small RE means that the input sample is normal. Thus, an anomaly is detected when the RE becomes large. Anomalous data can be detected when the RE exceeds a predefined threshold $\bar{RE}$ [45,46].

The input data and the output data are the same in standard AE training; however, by adding artificial noise only to the input data in the training phase, the output of the AE is expected to reproduce the denoised inputs. That is, the AE can be used for noise reduction, in which case, it is called a denoising autoencoder (DAE) [47].

#### 2.2.2. AE-Based Extrasystole Detection (AED)

Extrasystoles in the RRI data can be detected by using AE-based anomaly detection. This method is referred to as AE-based extrasystole detection (AED).

To train an AE for anomaly detection, a sufficient amount of normal RRI data with no extrasystoles has to be collected. The normal RRI data collected from the *i*th person $P_{i}$ is expressed as
(3)$$\begin{array}{c} x^{\left[i\right]}={\left[r_{1}^{\left[i\right]},\;r_{2}^{\left[i\right]},\;\cdots ,\;r_{j}^{\left[i\right]},\cdots ,\;r_{J_{i}}^{\left[i\right]}\right]}^{T} \end{array}$$
where $r_{j}^{\left[i\right]}$ denotes the *j*th RRI of $P_{i}$, and $J_{i}$ is the number of RRIs collected from $P_{i}$. The *i*th normal RRI data matrix, $X^{\left[i\right]}$ is constructed as follows:(4)$$\begin{array}{c} X^{\left[i\right]}=\left[\begin{array}{cccc} r_{1}^{\left[i\right]} & r_{2}^{\left[i\right]} & r_{3}^{\left[i\right]} & r_{4}^{\left[i\right]}\\ r_{2}^{\left[i\right]} & r_{3}^{\left[i\right]} & r_{4}^{\left[i\right]} & r_{L+1}^{\left[i\right]}\\ \vdots & \vdots & \vdots & \vdots \\ r_{J_{i}-3}^{\left[i\right]} & r_{J_{i}-2}^{\left[i\right]} & r_{J_{i}-1}^{\left[i\right]} & r_{J_{i}}^{\left[i\right]} \end{array}\right]. \end{array}$$

This type of matrix is called a Hankel matrix. The number of columns of $X^{\left[i\right]}$ was fixed to four in this study, which means that the number of input RRIs of the proposed AED was four. The reason for this setting is described in the following Section 2.2.3.

When normal RRI data from *I* persons $P_{1},\cdots ,P_{I}$ are collected, the normal RRI data matrix can be written as
(5)$$\begin{array}{c} X=\left[\begin{array}{c} X^{\left[1\right]}\\ X^{\left[2\right]}\\ \vdots \\ X^{\left[I\right]} \end{array}\right]. \end{array}$$

As preprocessing, each column in the normal RRI data matrix *X* is centered with a zero means. The AE is trained from the centered *X* by using the objective function Equation (Equation 1).

Since the trained AE detects any type of ectopic RRI data, the proposed AED classifies their types using conditional branches. The flowchart of the proposed AED is described in Figure 5.

We assumed that the AE was already trained from *X*. Before the AED starts, input RRI data with a window size of four RRIs have to be prepared in step (2). The *t*th piece of RRI data is described as follows:(6)$$\begin{array}{cc} x_{t}=[r_{t-3},\;r_{t-2},\; & r_{t-1},\;r_{t}]. \end{array}$$

In step (3), the process waits for the next (t+1)th RRI measurement because the (t+1)th RRI is needed to discriminate between PVA and PAC, as the difference between the two is the length of the RRI after arrhythmia occurrence. In step (4), the reconstruction error RE is calculated by using the trained AE. In step (5), $r_{t}$ is regarded as a normal sinus if the RE is less than its threshold $\bar{RE}$.

The conditional expressions in steps (6) and (7) are for evaluating whether $r_{t}$ is a PVC or R wave detection error since the pre-PVC RRI becomes short; the post-PVC RRI becomes long in PVC, and just one RRI becomes about double in length in a detection error. Here, ${\bar{r}}_{1}$ and ${\bar{r}}_{2}$ are the thresholds of $r_{t}$. If $r_{t}$ does not satisfy the conditional expression in step (8), short RRIs may occur successively. Such RRIs are neither PVC, PAC, nor R wave detection error and, thus, are categorized in this study as “other arrhythmias,” such as long QT syndrome, which healthy persons do not have. Step (9) checks the next (t+1)th RRI $r_{t+1}$ to discriminate between PVC and PAC. After the RRI classification ends, t=t+1 and the process returns to step (1).

In this procedure, the three thresholds $\bar{RE},{\bar{r}}_{1}$, and ${\bar{r}}_{2}$ need to be defined for adequate extrasystole detection. $\bar{RE}$ is defined as the maximum RE of the normal RRI data that were not used for training. ${\bar{r}}_{1}$ and ${\bar{r}}_{2}$ are tuning parameters; however, they can be determined as α% confidence limits. In other words, they are set so that α% of the samples representing the normal RRIs are below the control limits, and the other (100−α)% are above them. The control limits become large as α becomes large. Usually, the 99% confidence limit is adopted in anomaly detection techniques [48,49].

#### 2.2.3. DAE-Based Extrasystole Modification (DAEM)

The detected extrasystole is modified using DAE, which is referred to as DAE-based extrasystole modification (DAEM). We assumed that more than two successive extrasystoles do not occur because such successive extrasystoles rarely occur in healthy people [20]. Thus, we considered isolated extrasystoles.

In DAEM, multiple DAE models need to be trained to cope with different types of ectopic RRIs. In this research, two DAE models were constructed for PVC and PAC. To construct the training data for these DAE models, the normal RRI data have to be contaminated with artificial PVC or PAC since the DAE training requires both the normal RRI data *X* before containing PVC or PAC.

Thus, RRI data with real PVC or PAC are not used for DAE training.

The RRI data *X* with artificial PVC or PAC are denoted as $X_{\mathrm{PVC}}^{\prime }$, $X_{\mathrm{PAC}}^{\prime }$, respectively, and the training data consist of the output: *X* and the input: $X_{\mathrm{PVC}}^{\prime }$ or $X_{\mathrm{PAC}}^{\prime }$. The method for DAE training is the same as for AE except for the training data. These DAE models are referred to as ${\mathrm{DAE}}_{\mathrm{PVC}}$ and ${\mathrm{DAE}}_{\mathrm{PAC}}$.

With PVC, modifying only one RRI is not possible because PVC also changes the next RRI to a PVC occurrence. In PAC, the next several RRIs after the PAC occurrence need to be modified to compensate for the time gap between the real-time and the modified RRIs. In addition, since DAE requires inputs for precise noise reduction, RRIs other than the extrasystole may be slightly altered. The number of input RRIs of DAE should be small to prevent unnecessary RRI changes. Thus, the appropriate number of input RRIs should be determined for extrasystole modification. By considering these factors, the following input RRIs were used in any type of extrasystole modification:(7)$$\begin{array}{c} x=[r_{t-1},\;r_{t},\;r_{t_{+}1},\;r_{t+2}] \end{array}$$
where $r_{t}$ is an ectopic RRI detected by DAEM. Thus, the number of input RRIs in AED was determined as four.

The procedure of the proposed DAEM is described in Algorithm 1. We assumed that each of the three DAE models was already trained. In step (3), the newly measured *t*th RRI $r_{t}$ is applied to AE-ERD to classify it as normal, PVC, PAC, R wave detection error, or another type of arrhythmia. If $r_{t}$ is PVC or PAC, the process needs to wait for the (t+1)th and (t+2)th RRI to construct the input RRIs of DAE, *x*, in steps (13) and (14), and *x* is centered in step 15. In steps 16 and 17, the DAE model is loaded according to the classification result by means of AED and applied to the centered input $x^{\prime }$ in order to attain the output ${\hat{x}}^{\prime }$. In step 18, the output ${\hat{x}}^{\prime }$ is restored to $\hat{x}$ by adding the mean $\bar{x}$. There may be a difference in the time length between the original input RRI data *x* and the modified RRI $\hat{x}$ because the sum of the RRIs modified by DAE may not correspond with that of the original input RRIs. Thus, such a time length difference has to be compensated in steps 19 and 20.
**Algorithm 1** DAEM  1:  **while  do**  2:   Measure the *t*th RRI $r_{t}$.  3:   Apply AED to $r_{t}$.  4:   **if**
$r_{t}$ is normal. **then**  5:    t=t+1 and return to step 2.  6:   **else if**
$r_{t}$ is other types of arrhythmia. **then**  7:    Display “other types of arrhythmia.”  8:    t=t+1 and return to step 2.  9:   **else if**
$r_{t}$ is R wave detection error. **then**10:    Display “R wave detection error.”11:    t=t+1 and return to step 2.12:   **else**13:    Wait measurement of the t+1 and t+2 RRI $r_{t+1}$ and $r_{t+2}$.14:    Configure the input RRIs: $x=[r_{t-1}\;r_{t}\;r_{t_{+}1}\;r_{t+2}]$.15:    $x^{\prime }=x-\bar{x}$ where $\bar{x}$ is the mean of *x*.16:    Load either of DAE models: ${\mathrm{DAE}}_{\mathrm{PVC}}$ or ${\mathrm{DAE}}_{\mathrm{PAC}}$ according to the discriminated type of extrasystole by AED.17:    Input $x^{\prime }$ to the loaded DAE model and get the output ${\hat{x}}^{\prime }$.18:    $\hat{x}={\hat{x}}^{\prime }+\bar{x}$.19:    $d=\sum \hat{x}-\sum x$.20:    $\tilde{x}=\hat{x}-\left(d/4\right)1$.21:    Output $\tilde{x}$ as the modified RRI.22:    t=t+1 and return to step 2.23:   **end if**24:  **end while**

### 2.3. Data Description

Since the true RRI data before extrasystole occurs are unknown when real RRI data are used for evaluation, RRI data with artificial extrasystoles were used instead. This study used the MIT-BIH normal sinus rhythm database (NSRDB) [50,51] instead of the MIT-BIH Arrhythmia Database [52] because the ‘true’ RRI values were needed for modification performance evaluation

The NSRDB consists of the ECG and RRI data of eighteen healthy adult subjects A–R [50,51]. The subjects were five men aged 26–45 years (mean: 33.8 years, SD: 7.7 years) and thirteen women aged 20–50 years (mean: 35.8 years, SD: 7.7 years) who were diagnosed as healthy and did not have significant arrhythmias. A total of 166 datasets were constructed from eighteen subjects, and their total recorded length is about 375 h.

In this research, the subject data were organized into the following subgroups:Subject A: training data for AED.Subject B: training data for DAEM.Subjects C and D: parameter tuning data for AED.Subjects E and F: parameter tuning data for DAEM.Subjects G–L: test data without any ectopic RRIs.Subjects M–R: test data with ectopic RRIs.

The artificial extrasystole generation procedure assuming healthy persons is as follows:**PVC**: PVC alters both the pre-PVC RRI and the post-PVC RRI but usually does not affect other RRIs; the former RRI becomes short, and the latter RRI becomes long to compensate the heartbeat timing. To simulate a compensatory pause of PVC, artificial noise was added at random points, as shown in Figure 6 (left). The peak height of *H* was randomly set as 100 ms <H<370 ms so that the QT interval did not become shorter than the healthy QT interval [53]. In this research, we assumed that PVC on a T wave and successive PVCs did not occur because the target was a healthy person who rarely had successive extrasystoles.**PAC**: In PAC occurrence, only the former RRI becomes short, and heartbeat timing is not compensated. To simulate these characteristics, artificial noise was added at random points, which is shown in Figure 6 (right). The peak height of the artifact −H was randomly set between as 100 ms <H<370 ms so that the QT interval did not become shorter than the healthy QT interval [53]. We assumed that successive PACs did not occur because the target was a healthy person who rarely had successive extrasystoles.

These extrasystoles were generated at a rate of one per about 1200 beats, which means that an extrasystole occurred about 70–80 times per day since even healthy persons may have about ten to one hundred extrasystoles a day. The points where artifacts were added were recorded for the evaluation of extrasystole detection.

For AED training, 500 samples were randomly clipped from the RRI data of subjects A and B, and their total length was about 23 min and 33 min, respectively.

## 3. Results

### 3.1. Performance Evaluation

This study adopted the sensitivity (SEN) [%] and false positive (FR) rate [times/hour] to evaluate the performance of the extrasystole detection of AED. In addition, the root mean squared error (RMSE) and its decrement rate $C_{\mathrm{RMSE}}$ were used to evaluate the extrasystole modification performance, which are defined as follows:(8)$$\begin{array}{c} \mathrm{RMSE}=\sqrt{\frac{1}{N}\sum_{k=1}^{N}{\left(y_{i}-{\hat{y}}_{i}\right)}^{2}} \end{array}$$(9)$$\begin{array}{c} C_{\mathrm{RMSE}}=\frac{{\mathrm{RMSE}}_{\mathrm{modified}}}{{\mathrm{RMSE}}_{\mathrm{ectopic}}}\times 100\;\left[\%\right] \end{array}$$
where *N* is the number of samples, and $y_{i}$ and ${\hat{y}}_{i}$ denote the *i*th reference and estimate. ${\mathrm{RMSE}}_{\mathrm{extrasystole}}$ and ${\mathrm{RMSE}}_{\mathrm{modified}}$ are the RMSE between the original RRI data and the artificial extrasystole RRI data, and the RMSE between the artificial extrasystole RRI data and the RRI data modified by the proposed DAEM.

The simulation procedure—artificial extrasystole data generation, AE and DAE training, and extrasystole detection and modification by AED and DAEM—was repeated ten times independently for precise performance evaluation.

### 3.2. Extrasystole Detection

The datasets of subject A for AED training were used and those of subjects C and D were used for parameter tuning. The number of units in the hidden layer in the AE model was three. A sigmoid function and an identity function were adopted as the activation functions in the hidden and output layers.

The RRI data collected from subjects G–R were used to test the extrasystole detection of the proposed AED. The application result of AED showed that no FP occurred in subjects G–L whose data did not include any extrasystoles, as well as showing a sensitivity of 93% and an FP rate of 0.08 [times/hour] in the subjects M–R who had extrasystoles. The accuracy of the ectopic type classification was 96%. This result clearly shows that the proposed AED functioned successfully.

### 3.3. PVC Modification

In ${\mathrm{DAE}}_{\mathrm{PVC}}$ training, the datasets of subject B and those of subjects E and F were used for training and parameter tuning. The number of units in the hidden layer was two. The activation functions in the hidden layer and the output layer were a rectified linear unit (ReLU) and an identity function.

Figure 7 shows an example of an application result of the proposed DAEM to the RRI data with artificial PVC. The average decrement rate of the RMSE was 31%.

Figure 8 illustrates the HRV features that were extracted from the RRI data with PVC and the RRI data modified by DAEM shown in Figure 7. Their average decrement rates of RMSE are summarized in Table 1 suggests the usefulness of the proposed DAEM for the RRI modification when PVC occurs.

### 3.4. PAC Modification

The datasets of subject B used for the training of ${\mathrm{DAE}}_{PAC}$ and those of subjects E and F were used for parameter tuning so that the reconstruction performance of ${\mathrm{DAE}}_{PAC}$ was maximized. The number of units in the hidden layer was two. The activation functions in the hidden and output layers were a ReLU and an identity function, respectively.

Figure 9 shows the results of applying the proposed DAEM to the RRI data with artificial PAC, and the average decrement rate of RMSE was 73%.

HRV features were extracted from the RRI data with PAC, and the modified RRI data shown in Figure 9 are illustrated in Figure 10. Their average decrement rates of RMSE are summarized in Table 1. Thus, the influence of PAC on the HRV analysis was suppressed by the proposed DAEM.

## 4. Discussion

The sensitivity and FP rate of the proposed AED were 93% and 0.08 times/hour, and the ectopic type classification accuracy was 96%. These results indicate that the AED functioned well and that the RRI data collected from only one healthy person were enough for the AE training since the ectopic RRIs caused by extrasystoles are entirely different from normal RRIs.

Although previous studies used ECG signals for extrasystole detection, the proposed method does not analyze the ECG signals but the RRI data. Thus, the proposed AED was compared with another anomaly detection method, singular spectrum analysis (SSA), which is an anomaly detection method used for time series data [54]. The anomaly detection model of SSA is derived through the singular value decomposition (SVD) of a Hankel matrix Equation (Equation 4), which means that SSA is a liner model. SSA achieved a sensitivity of 95% and an FP rate of 1.2 times/hour, which is worse than the AE. Since the AE was based on a neural network that could express nonlinearity, and HRV is essentially a nonlinear phenomenon [2], AE was more appropriate than SSA for extrasystole detection. On the other hand, this study did not compare the extrasystole detection methods mentioned in Section 1 [26,27,28] since they analyzed raw ECG signals.

The RMSE decrement rate of PVC modification was 31%, and that of PAC modification was 73%, which indicates that it was more difficult for PAC to be modified adequately than PVC. As PAC does not have a compensatory pause, unlike PVC, PAC modification is required for synchronization with real-time data. Some RRIs around the PAC must be modified as well as the RRI directly affected by PAC so that the time gap is compensated. Thus, the PAC modification performance did not reach that of PVC modification.

The effect of DAEM on the normal RRI data was checked. It is possible that the normal RRI data were incorrectly modified since a few false positives occurred in the extrasystole detection, although the rate was only 0.08 times/hour. The result of applying DAEM to the normal RRI data showed that the average alternation width of RRI was less than 2.4 ms. This value is acceptable for HRV analysis because ECG should be measured with at least 200 Hz for precise HRV analysis according to the HRV analysis guideline [2]. Thus, the RRI measurement error of within 5 ms is acceptable for the clinical application of HRV analysis.

The modification performance of the DAE was compared with regression methods; partial least squares (PLS) and locally weighted PLS (LWPLS) were considered here because Kamata et al. attemtped R wave detection error modification using PLS and LWPLS that did not use ECG signals but RRI data [35]. PLS is a widely-used linear regression method that can build an accurate model with a small number of latent variables. LWPLS is an expansion of PLS based on the framework of Just-In-Time (JIT) [55] modeling for dealing with nonlinearity and system characteristics change [56,57]. In LWPLS, a local PLS model is built using weighted samples stored in a database according to the similarity between the query and the weighted samples only when an estimate is requested. The constructed local model represents a nonlinear relationship between the input and the output around the query because a nonlinear relationship can be approximated as a linear relationship in a small region. The local model is purged after being used for estimation. Figure 11 shows the RMSE decrement rates of the RRI in PVC modification by DAE, PLS, and LWPLS as evaluated through ten independent calculations. These box plots show that DAE achieved the best performance among the three methods. The same tendency was confirmed in the PAC modification.

The RRI fluctuations caused by PVC or PAC are a highly nonlinear phenomenon because the RRI fluctuation width is random. Since PLS and LWPLS are linear methods, it is difficult to recover RRI fluctuations sufficiently. On the other hand, the proposed method adopted DAE, which can cope with nonlinearity well, and improved the RRI modification performance in comparison with PLS and LWPLS.

In addition, the proposed DAEM was trained with R wave detection error modification. The artificial ectopic RRI that occurred due to the R wave detection error was generated as
(10)$$\begin{array}{c} r_{j}={\tilde{r}}_{j}+{\tilde{r}}_{j+1} \end{array}$$
where $r_{j}$ is the measured *j*th RRI [ms] and ${\tilde{r}}_{j}$ denotes the *j*th “true” RRI measurement if both the *j*th and (j+1)th R waves were detected properly. We assumed that successive detection errors did not occur [35]. The DAE model for the R wave detection error modification ${\mathrm{DAE}}_{\mathrm{RDE}}$ was trained using the datasets of subject B, and its parameters were tuned with the datasets of subjects E and F. The number of units in the hidden layer became eight. A sigmoid function and an identity function were adopted as the activation functions in the hidden and output layers.

The application result of DAEM to R wave detection error was compared with PLS and LWPLS, which is shown in Figure 12. The RMSE decrement rate of RRI could not be calculated since the number of RRIs was altered before and after the modification. Thus, we evaluated the RMSE between the original RRI data before the artificial detection error and modified RRI data. Figure 12 shows that the average RRI improvements were almost the same among the three methods; however, the deviation of DAE was smaller than that of the other two, which means that it was possible to construct a stable model with DAE. Although a single detection error was considered in this work, Kamata et al. described a method for dealing with successive detection errors [35], which modified successive detection errors step by step. This method can also be used for the proposed DAEM when successive detection errors are modified.

This study applied the proposed methodologies to another type of arrhythmia that healthy persons never have. In this research, artificial RRI data containing atrial fibrillation (AF) was considered. In AF, rapid and irregular beating of the atrial chambers of the heart occurs [58]. One-minute AF data were generated by adding random numbers following a uniform distribution between −50 ms and 50 ms to the constant RRI values. Figure 13 shows an example of RRI data containing AF. AED did not detect almost any AF. Since the variation width of the RRIs in AF is smaller than in PVC and PAC, the proposed AED might not detect AF. However, this study can adopt existing RRI-based AF detection methods [59,60] when patients with an AF risk are monitored.

The proposed extrasystole treatment framework was applied to a real health monitoring problem. Fujiwara et al. developed an epileptic seizure prediction algorithm by combining HRV analysis and an anomaly detection algorithm referred to as multivariate statistical process control (MSPC). Since extrasystole occurrences may affect HRV and cause false positives in seizure prediction, they should be detected and modified appropriately. Figure 14 (top) shows interictal RRI data with PVC measured from a patient with left mesial temporal lobe epilepsy (female, 31 y.o.), and the enlarged ECG data around the PVC occurrence is shown in Figure 14 (bottom), in which R waves were detected by means of a first derivative-based peak detection algorithm. This data was collected from the Tokyo Medical and Dental University (TMDU) hospital. The retrospective evaluation of clinically acquired data was approved by the Medical Research Ethics Committee of the TMDU hospital. The details of the clinical data used in this research are described in [19].

In the seizure prediction algorithm, the abnormality indexes and the $T^{2}$ and *Q* statistics are calculated by MSPC, and seizures are predicted when either the $T^{2}$ or *Q* statistic exceeds the predefined control limits for ten seconds [19]. The blue lines in Figure 15 are the $T^{2}$ and *Q* statistics derived from the original RRI data with PVC, and the horizontal lines denote the control limits. This figure shows that a false positive occurred around 350 s in the *Q* statistic, which corresponded with the PVC occurrence in Figure 14. We confirmed that this false positive did not reflect any epileptic EEG discharge according to the EEG data and that it was only caused by the PVC. Thus, the PVC had to be detected and modified appropriately to suppress false positives.

This study attempted PVC detection and modification using the proposed extrasystole treatment framework. AED correctly detected the PVC, and the DAEM modified the detected PVC, the result of which is shown in Figure 14 as the red line. The $T^{2}$ and *Q* statistics were derived from the modified RRI data. The red line in Figure 15 (bottom) indicates that the false positive in the *Q* statistic did not occur by modifying the PVC. This result clearly shows that the proposed method correctly prevented the false positive caused by PVC, which is of importance for the quality of life improvement of patients with epilepsy.

Therefore, the proposed extrasystole treatment framework contributes to realizing highly-adequate HRV-based health monitoring services.

## 5. Conclusions

In this study, we proposed an extrasystole treatment framework based on neural networks, in which extrasystole occurrences are detected with AE (AED) and are modified with DAE (DAEM). The proposed framework can deal with ectopic RRIs caused by PVC and PAC. The case study showed that the sensitivity and the FP rate of extrasystole detection were 93% and 0.08 times per hour, and the accuracy of the ectopic type classification was 96%. The extrasystole modification performances of the proposed DAEM were RRI improvement rates of 31% and 73% in PVC and PAC, respectively. The usefulness of the proposed framework was demonstrated through its application to the real health monitoring problem: the proposed extrasystole treatment framework was able to improve the epileptic seizure prediction performance. Thus, the proposed framework can contribute to realizing accurate HRV-based health monitoring and medical sensing systems.

The limitations of this study include the assumption that only PVC or PAC occurring in healthy persons were considered and that frequent morbid extrasystoles do not occur. In healthy persons, this assumption is correct, and the proposed methodologies cover the majority. The RRI data with real extrasystoles could not be used; rather, RRI data with artificial extrasystoles was used for quantitative evaluation. In future work, other types of arrhythmia and successive arrhythmia will be included to expand the target of the proposed methodologies.

Embedded software with the proposed framework is being developed for the microcomputer of a wearable heart rate sensor. The sensor under development may contribute to realizing precise HRV-based health monitoring services since it would allow robust HRV analysis.

## Figures and Tables

**Figure 1 sensors-21-03235-f001:**
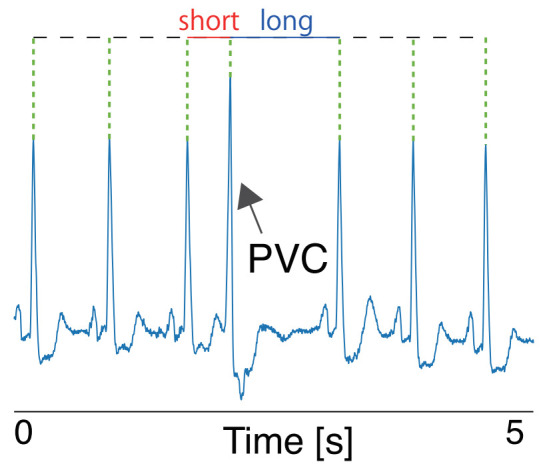
Example of an ECG with a PVC. The RRIs both before and after are altered in most PVC occurrences: RRIs before the PVC become short, and the RRIs after the PVC become long to compensate for the heartbeat rhythm. This phenomenon is called a compensatory pause.

**Figure 2 sensors-21-03235-f002:**
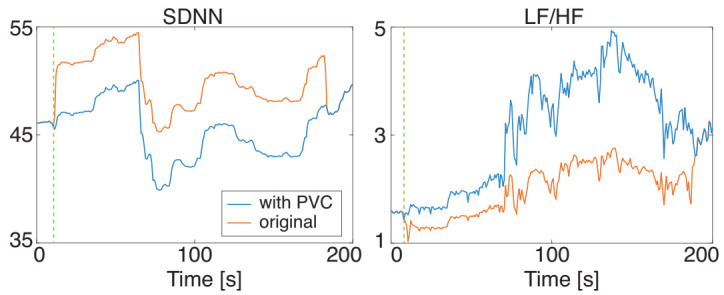
Influence of PVC on HRV features: SDNN (**left**) and LF/HF (**right**): The blue and red lines show the features extracted from the RRI data without PVC and with PVC, respectively. The vertical dashed line denotes the point of the PVC occurrence.

**Figure 3 sensors-21-03235-f003:**
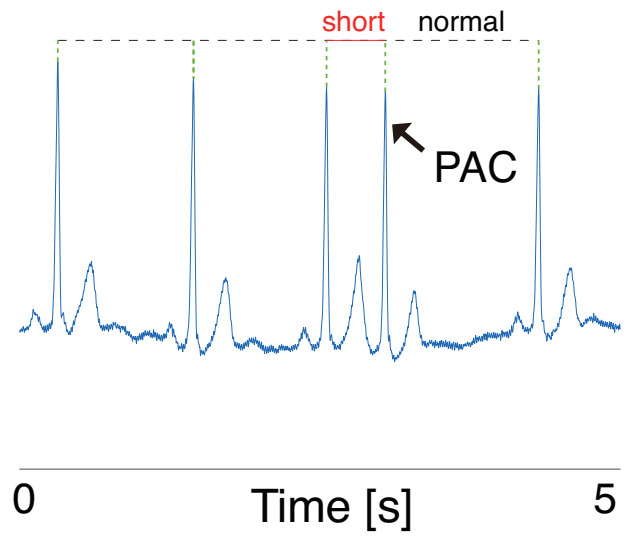
Example of an ECG with PAC. As the sinus rhythm resets after the PAC occurrence, only one RRI before PAC becomes short. That is, PAC does not have a compensatory pause.

**Figure 4 sensors-21-03235-f004:**
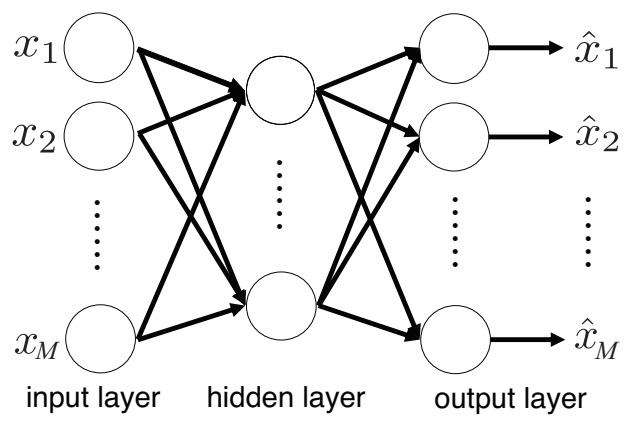
Structure of an AE. An AE consists of an input layer, a hidden layer, and an output layer. $x_{1},\cdots ,x_{M}$ and ${\hat{x}}_{1},\cdots ,{\hat{x}}_{M}$ are the input variables and output variables of the AE, respectively. Circles denote units that express activation functions.

**Figure 5 sensors-21-03235-f005:**
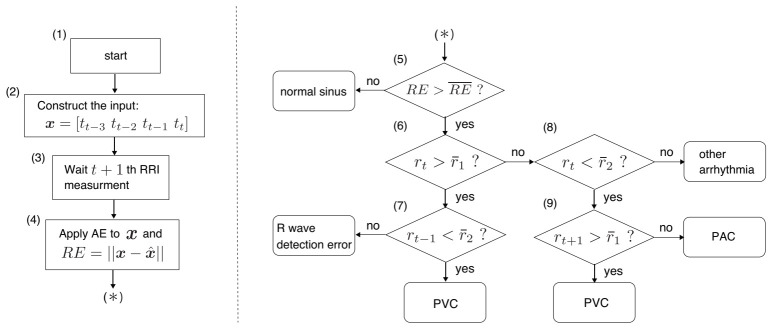
Flowchart of AED. This procedure can classify the RRI data into normal sinus, PVC, PAC, R wave detection error, or another type of arrhythmia.

**Figure 6 sensors-21-03235-f006:**
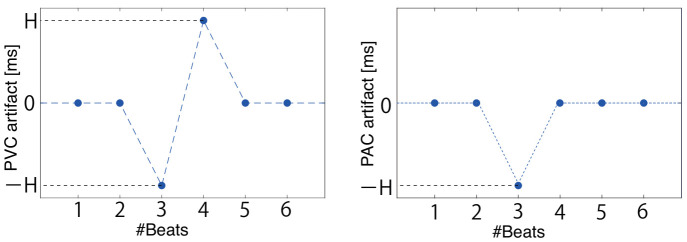
Artificial noise for generating PVC (**left**) and PAC (**right**), which was added at random points in the RRI data. The peak height of *H* was randomly set as 100 ms <H<370 in this study.

**Figure 7 sensors-21-03235-f007:**
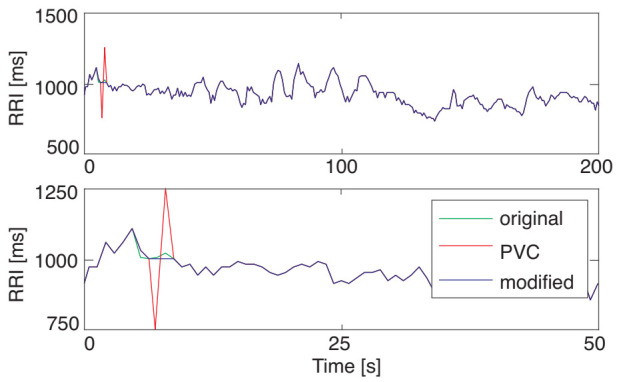
The RRI modification result of PVC (**top**) and its enlarged display (**bottom**). The blue, red, and green lines represent the original RRI data, the RRI data with artificial PVC data, and the RRI data modified by the proposed DAEM, respectively. The modified data (the blue line) overlapped almost entirely with the original data (the green line) due to the high modification performance.

**Figure 8 sensors-21-03235-f008:**
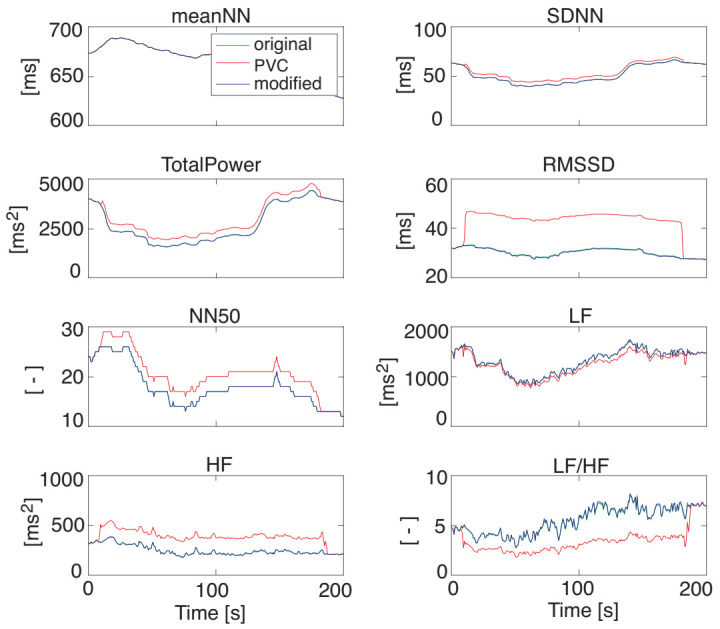
The HRV features extracted from the RRI data with and without PVC modification. The blue, red, and green lines represent the original HRV data, the HRV data extracted from the RRI with artificial PVC data, and the HRV data extracted from the RRI data modified by the proposed DAEM, respectively. The modified data (the blue line) overlapped almost entirely with the original data (the green line).

**Figure 9 sensors-21-03235-f009:**
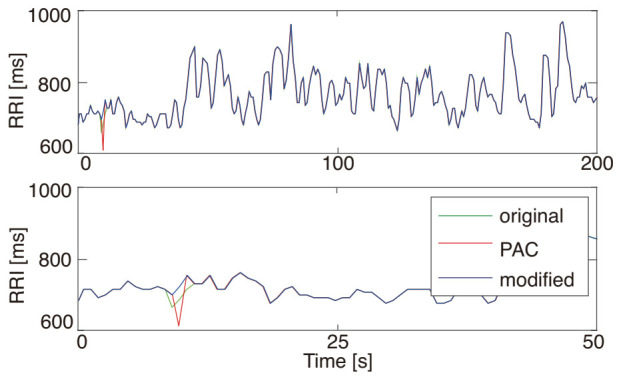
The RRI modification result of PAC. The blue, red, and green lines represent the original RRI data, the RRI data with artificial PVC data, and the RRI data modified by DAEM, respectively. This figure shows that the modified RRI was very close to the original RRI data.

**Figure 10 sensors-21-03235-f010:**
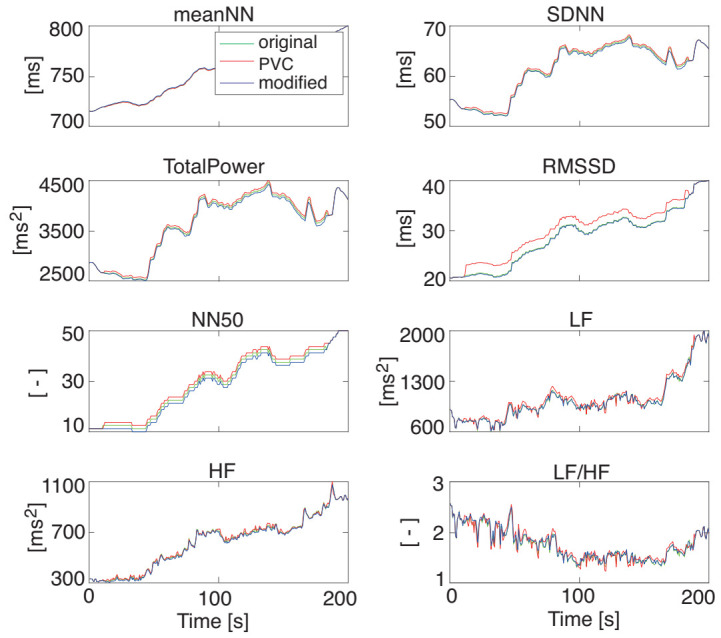
The HRV features extracted from the RRI data with and without PAC modification. The blue, red, and green lines represent the original HRV data, the HRV data extracted from the RRI with artificial PAC data, and the HRV data extracted from the RRI data modified by the proposed DAEM, respectively. This figure shows that the HRV data extracted from the modified RRI became close to the HRV data extracted from the original RRI data.

**Figure 11 sensors-21-03235-f011:**
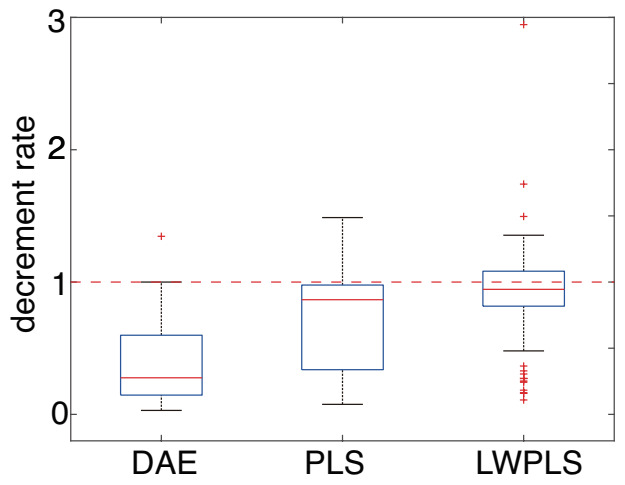
The RMSE decrement rates of the RRI data with PVC by DAE, PLS, and LWPLS. The proposed DAE achieved the best performance among the three methods.

**Figure 12 sensors-21-03235-f012:**
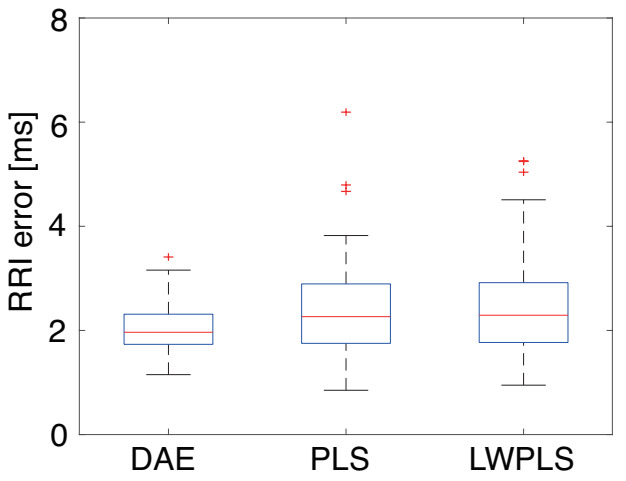
The RRI errors in the R wave detection error modification. Although the average RRI improvements were almost the same, the deviation of DAE was smaller than that of the other two.

**Figure 13 sensors-21-03235-f013:**
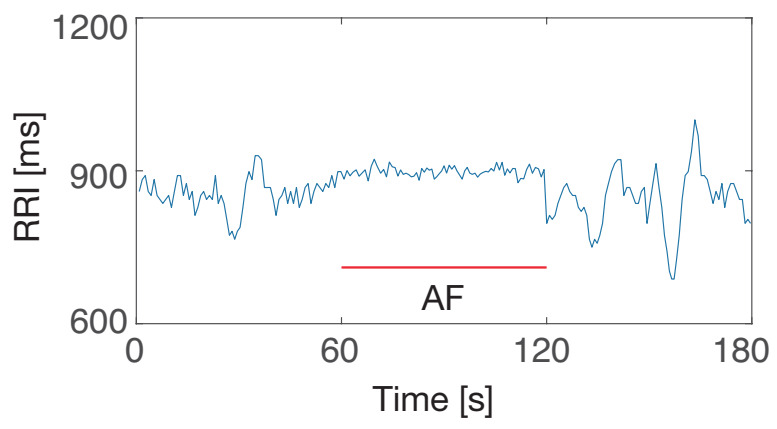
An example of artificial RRI data containing AF.

**Figure 14 sensors-21-03235-f014:**
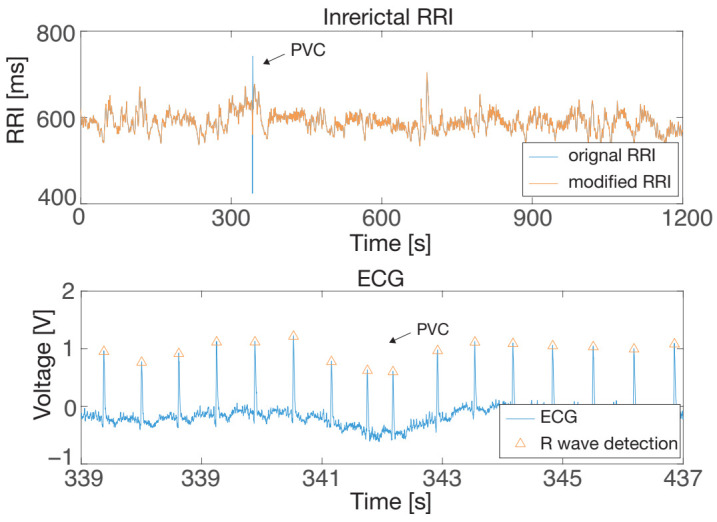
An interictal episode collected from an epileptic patient: RRI data with PVC (**top**) and the ECG data enlarged around the PVC (**bottom**). The arrow symbol denotes the PVC occurrence.

**Figure 15 sensors-21-03235-f015:**
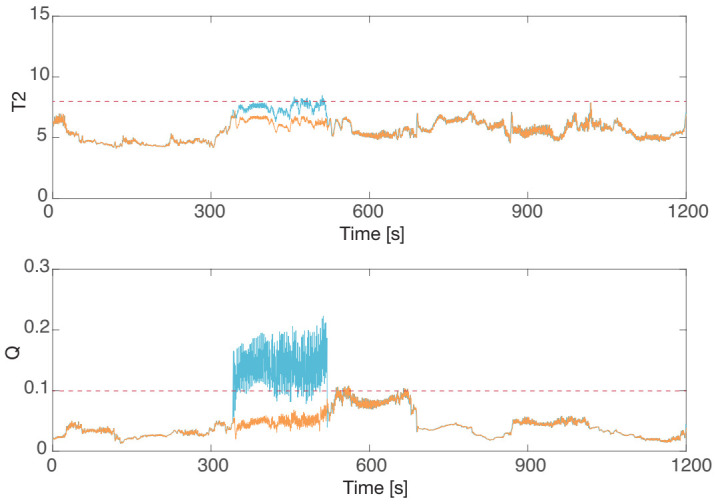
The epileptic prediction result of an interictal episode by the $T^{2}$ (**top**) and *Q* (**bottom**) statistics: the original RRI data (blue) and modified RRI data (red). The false positive caused by PVC was successfully suppressed using the proposed DAEM.

**Table 1 sensors-21-03235-t001:** The average decrement rates [%] of RMSE by DAEM.

	PVC	PAC
RRI	31	73
meanNN	45	90
SDNN	12	45
Total Power	13	77
RMSSD	4	29
NN50	20	66
LF	31	77
HF	26	72
LF/HF	27	70

## Data Availability

The clinical data will be made available by the corresponding author to colleagues who propose a reasonable scientific request after approval by the Medical Research Ethics Committee of TMDU.

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
