# Peer review of "Autoencoder-Based Extrasystole Detection and Modification of RRI Data for Precise Heart Rate Variability Analysis"

_sensors, 2021, doi:10.3390/s21093235_

Round 1

Reviewer 1 Report

The paper discussed a framework for  extrasystole occurrence detection from the RRI data. The paper is well written in a good English. I have two major comments:

  • The link with the topics of Sensors journals should be better explained. Data comes from senzors. However this should be emphasized.
  • Second comparison with existing algorithms showing how the proposed algorithm performs better should be stated more clearly. Please add on this

Given the two major comments I recommend a major revission

Author Response

The paper discussed a framework for extrasystole occurrence detection from the RRI data. The paper is well written in a good English. I have two major comments:

Thank you for your good evaluation. In this revision, we modified our manuscript following your comments. Our replies are as follows:

The link with the topics of Sensors journals should be better explained. Data comes from senzors. However this should be emphasized.

We agree with your indication. Our original manuscript rather focused on only software and machine learning aspects. We added the following description about the connection with sensors in Sec. 1.

“This problem should be solved with a software side after ECG measurement; RRI data collected from ECG sensors have to be checked and appropriately treated by software before extracting HRV when there is the possibility that ectopic RRIs are contained.”

In addition, we emphasized the connection with sensors in Sec. 5.

“We are currently developing embedded software with the proposed framework for a microcomputer of a wearable heart rate sensor. The sensor under development may contribute to realizing precise HRV-based health monitoring services since it would allow robust HRV analysis.”

Second comparison with existing algorithms showing how the proposed algorithm performs better should be stated more clearly. Please add on this.

Thank you for your important comment. The existing algorithm is based on partial least squares (PLS) which is a kind of linear regression method; however, the RRI fluctuations caused by PVC or PAC are highly nonlinear phenomena because such RRI fluctuation width is random. Thus, it is difficult for linear regression to recover RRI fluctuations. On the other hand, we adopted an autoencoder that can handle nonlinearity well and improved the RRI modification performance in comparison with the existing method. 

We added the above description in Sec. 4 of the revised manuscript as follows:

“The RRI fluctuations caused by PVC or PAC are highly nonlinear phenomenon because RRI fluctuation width is random. Since PLS and LWPLS are linear methods, it is difficult to recover RRI fluctuations sufficiently. On the other hand, the proposed method adopted DAE that can cope with nonlinearity well and improved the RRI modification performance in comparison with PLS and LWPLS.”

Reviewer 2 Report

The paper proposes the extrasystole treatment framework based on neural networks. Extrasystole occurrences are detected with autoencoder (autoencoder-based extrasystole detection) and modified with denoising autoencoder (denoising autoencoder-based extrasystole modification).

The paper is well organized, and the length is appropriate. The title is chosen correctly, and the abstract provide sufficient information to give a clear idea of what to expect from the paper.

The technical depth of the paper meets the requirements for a scientific article published in a quality journal.

Usually, abbreviations and acronyms are defined the first time they are used in the text. On line 10, the authors use the acronym ”DAE” which is defined only on line 91. Please do this small correction.

Author Response

The paper is well organized, and the length is appropriate. The title is chosen correctly, and the abstract provide sufficient information to give a clear idea of what to expect from the paper. The technical depth of the paper meets the requirements for a scientific article published in a quality journal.

We appreciate your positive evaluation. 

Usually, abbreviations and acronyms are defined the first time they are used in the text. On line 10, the authors use the acronym ”DAE” which is defined only on line 91. Please do this small correction.

Thank you for your indication. We corrected this error in this revision.

Reviewer 3 Report

The authors propose a novel extrasystole treatment framework that can deal with ectopic R-R intervals caused by extrasystoles–premature ventricular contraction (PVC) and premature atrial contraction (PAC). The manuscript is well structured, the methodology exposed seems appropriate and the results presented are really promising. As minor concerns:

  • The use of first persons (i.e., “we”, “their”, possessives, and so on) should be avoided and can preferably be expressed by the passive voice or other ways.
  • An important reference is missing in line 110 to support such claims.
  • In Figure 3 it should be set 5 s instead of 5000 ms. In Figure 9 the RR units could be set to seconds instead of ms. What does b] refer to? The vertical axis units in Figure 10 are missing and the units could be adjusted, for example, Total Power. The voltage units in Figure 14 should be also adjusted to V instead of mV.
  • The average decrement rates of RMSE exposed in lines 299 and 311 should be presented in a table to compare the results obtained with the PVC and PAC methods.

Author Response

The authors propose a novel extrasystole treatment framework that can deal with ectopic R-R intervals caused by extrasystoles–premature ventricular contraction (PVC) and premature atrial contraction (PAC). The manuscript is well structured, the methodology exposed seems appropriate and the results presented are really promising. As minor concerns:

Thank you for your good evaluation. We revised our manuscript following your comments. Our replies are as follows: 

The use of first persons (i.e., “we”, “their”, possessives, and so on) should be avoided and can preferably be expressed by the passive voice or other ways.

We modified the whole of our manuscript so that first persons were not included. 

An important reference is missing in line 110 to support such claims.

Thank you for your important suggestion. We added references about extrasystole in line 110.

In Figure 3 it should be set 5 s instead of 5000 ms. In Figure 9 the RR units could be set to seconds instead of ms. What does b] refer to? The vertical axis units in Figure 10 are missing and the units could be adjusted, for example, Total Power. The voltage units in Figure 14 should be also adjusted to V instead of mV.

Thank you for your indications about figures. We corrected these errors in this revision.

The average decrement rates of RMSE exposed in lines 299 and 311 should be presented in a table to compare the results obtained with the PVC and PAC methods.

We appreciate your valuable comment. We summarized these average decrement rates of RMSE into Table 1.

Round 2

Reviewer 1 Report

The current version is an improvement as compared with the previous version. However, still the link with the Sensors topic is still not completely well explain. I will give an accept with the recommendation that editors checks once more this issue.